computational biology/cellular biology

oncogenesis, non-cell-autonomous, model

**Author for correspondence:**
Alessandro Esposito
email: ae275@cam.ac.uk

# Cooperation of partially transformed clones: an invisible force behind the early stages of carcinogenesis

## Alessandro Esposito

The Medical Research Council Cancer Unit, University of Cambridge, Hills Road, Cambridge CB2 0XZ, UK

 AE, 0000-0002-5051-091X

Most tumours exhibit significant heterogeneity and are best described as communities of cellular populations competing for resources. Growing experimental evidence also suggests that cooperation between cancer clones is important as well for the maintenance of tumour heterogeneity and tumour progression. However, a role for cell communication during the earliest steps in oncogenesis is not well characterized despite its vital importance in normal tissue and clinically manifest tumours. Here, we present a simple analytical model and stochastic lattice-based simulations to study how the interaction between the mutational process and cell-to-cell communication in three-dimensional tissue architecture might contribute to shape early oncogenesis. We show that non-cell-autonomous mechanisms of carcinogenesis could support and accelerate pre-cancerous clonal expansion through the cooperation of different, non- or partially transformed mutants. We predict the existence of a 'cell-autonomous time horizon', a time before which cooperation between cell-to-cell communication and DNA mutations might be one of the most fundamental forces shaping the early stages of oncogenesis. The understanding of this process could shed new light on the mechanisms leading to clinically manifest cancers.

## 1. Introduction

The cooperation between tumour cells and their environment and the competition between different tumour clones during carcinogenesis are well established [1]. Other types of cooperation, for instance, the positive cooperation between tumour clones, or even non-transformed clones, have been increasingly recognized as a possible fundamental driving force in cancer as well [2,3]. The complexity of all possible clonal interactions, particularly during the late stages of cancer, is

therefore fostering research aimed to model cancer from an ecological perspective [1,2,4]. Competition for resources is one of the driving forces for clonal interaction. However, cell-to-cell communication is an equally fundamental mechanism mediating the interaction of cellular populations through shared diffusible or immobile molecules, such as cytokine or metabolites. After early modelling work on angiogenesis [2], the possibility that partially transformed tumour cells might cooperate was generalized by Axelrod *et al.* [3]. Several recent experimental findings are now supporting the notion that cooperation of clones and poly-clonality play an important role in the emergence of cancer.

Glioblastoma multiforme tumours, for instance, exhibit considerable intra-tumoural heterogeneity including the pathogenic expression of an oncogenic truncation of the epidermal growth factor receptor (ΔEGFR) gene and EGFR amplification [5]. The less frequent ΔEGFR clones can support an increased fitness of the more prevalent cells overexpressing EGFR, through secretion of IL6 and LIF and a paracrine effect. Recently, Reeves and colleagues [6] have used multi-colour lineage tracing with a Confetti mouse line together with the topical administration of a carcinogen, to study clonal evolution during early oncogenesis. Interestingly, the authors observed benign papillomas harbouring an HRAS Q61L mutation with streaks of Notch mutant clones. Although these Notch mutants were considered infiltrating clones with no active role in the oncogenic process, Janiszewska & Polyak [7] noted that cooperation between the Notch and HRAS mutants could not be excluded and that streaks of Notch clones are reminiscent of structures found in non-mutualistic colonies of budding yeast. Although unproven, it is conceivable that the less frequent clones can provide, altruistically, a fitness advantage to the HRAS mutant cells similarly to what has been observed for glioblastoma multiforme [5,8] or for WNT-secreting wild-type HRAS clones supporting HRAS mutants [9]. While facilitating the oncogenic process, a non-mutualistic clone would be then outcompeted by more aggressive clones after a clonal sweep and diversification into multiple intermixed mutants [6] suggestive of mutualistic clonal interactions [7].

However, it is unclear if these observations, often obtained using model systems with carcinogens or established tumour clones, can be recapitulated at the low-mutational rates occurring naturally [10]. Furthermore, it is unknown at which stage of carcinogenesis, non-cell-autonomous mechanisms might have a role [9]. As cell-to-cell communication and clonal interaction are often neglected in formal models of carcinogenesis, we propose a model for the interaction between the mutagenic process and cell-to-cell communication within a three-dimensional tissue architecture. We developed the simplest possible analytical models and test them with stochastic lattice-based simulations [2,11–15] and discrete-time Markov chain modelling [16,17] to capture the basic emergent properties of early oncogenesis in the presence of mutations and clonal cell-to-cell communication. We propose that the extremely low-mutational frequency encountered in physiological conditions does not render cooperation between mutations in adjacent cells unlikely but—rather the opposite—that synergy between the mutational process and cell-to-cell communication might play a fundamental role in carcinogenesis.

# 2. Results

## 2.1. A model for mutationally driven cooperation in oncogenesis

The question addressed in this work is not *if* cooperation between mutant (partially transformed) cells can occur, but *how likely* or *when* distinct mutations can occur in different cells cohabiting within the same tissue. Therefore, we develop a simple mathematical model to gain insights into answering these fundamental questions. We consider a low-mutational rate $\rho_0$, constant throughout oncogenesis and equal for each possible oncogenic mutation [18]. With oncogenic mutation, we refer to any mutation that at any given time (not necessarily when it occurs) might contribute to the increased fitness of a clone that will eventually evolve into cancer, either through cell-autonomous or non-cell-autonomous mechanisms.

The probability for a single cell to accrue two specific mutations independently within a given time interval is thus $p_0^2$ (with $p_0 = \rho_0 t \ll 1$). The probability that two neighbouring cells exhibit one given mutation each independently is, unsurprisingly, the same. Initially, we assume non-dividing cells in a well-organized tissue that after accumulating these two mutations acquire a fitness advantage. We will refer to these cells as initiated or transformed, but we will use these terms very loosely only to indicate a gain in fitness.

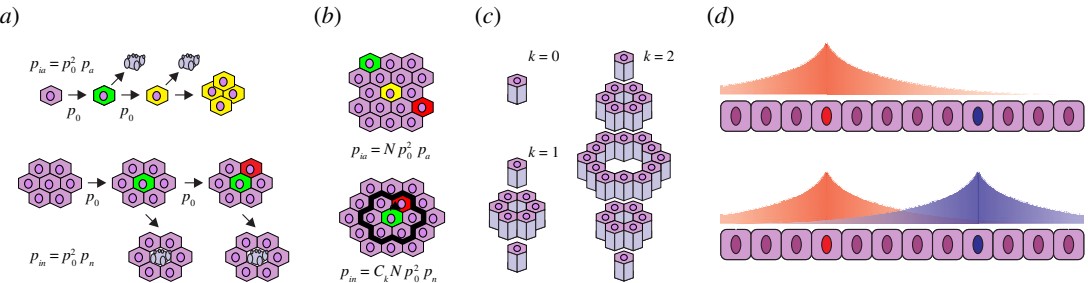

**Figure 1.** Tissue organization and non-cell-autonomous mechanisms. (*a*) A simple model where cells accumulate two mutations or two mutations occur in different cells within the same neighbourhood. (*b*) When the probability of accruing mutations is low, within a tissue of $N$ cells, there will be more opportunities for mutations to co-occur within a given neighbourhood rather than within the same cell. (*c*) The neighbourhood of a cell can be described as a problem of geometrical tessellation of space which will depend on tissue organization, here shown a simple example of hexagonal pillars tesselating space. (*d*) Gradients of shared resources (e.g. growth factors or metabolites) might be then induced by either one or the other cell triggering interactions by juxtacrine or paracrine effects.

In a tissue with $N$ cells, the probability of cell-autonomous initiation of one mutant cell is simply $p_{ia} = Np_0^2p_a$ Similarly, the probability of non-cell-autonomous initiation is $p_{in} = NCp_0^2p_n$. $p_a$ and $p_n$ are defined as the probabilities that one cell harbouring the right pair of mutations—either by itself or within its neighbourhood—survives tumour-suppressive mechanisms (figure 1*a*). $C$ is a coordination number, i.e. the number of cells within the neighbourhood of a reference cell (figure 1*b*,*c*). Within the validity of common assumptions (e.g. equally probable, spatio-temporally invariant and independent mutational events), the probability of initiation within a group of $N$ cells is the sum of $p_{ia}$ and $p_{in}$

$$p_i = Np_0^2p_a\left(1 + \frac{\Omega p_{n0}}{p_a}\right), \tag{2.1}$$

with $\Omega = Cp_n/p_{n0}$ and where $p_{n0}$ is the probability that one cell is transformed when directly in contact with another mutant. $p_n$ (and thus $\Omega$) depends on tissue organization and the type of cell-to-cell cue that contributes to the process of transformation (figure 1*d*). With this simple notation, the answer to our central question can be thus separated into the study of tissue organization (the factor $\Omega$) and the magnitude of $p_{n0}$ compared with $p_a$.

To model the organization of tissue wherein mutated cells are resident, several aspects of tissue organization have to be considered:

- (i) the more distant a neighbouring cell is, the lower the probability of cooperative non-cell-autonomous effects should be, i.e. $p_n$ shall be a function of distance ($d$);
- (ii) $C$ is the sum of cells in extended neighbourhoods or the sum of $C_k$, i.e. the number of cells in the $k$-neighbourhood (at a distance $d_k$), where $k = 1$ defines cells in contact (i.e. $d_1 = 0$);
- (iii) $C_k$ depends on tissue architecture that we model as a problem of three-dimensional tessellations of space;
- (iv) tissues are compartmentalized and, therefore, boundary effects should be considered.

Therefore, in general, the factor $\Omega$ can be described as the cumulative effect on the probability of initiation of a reference cell from each cell within a tissue as a consequence of a cell-to-cell communication

$$\Omega = p_{n0}^{-1}\sum_k C_kp_{nk}(d_k). \tag{2.2}$$

For convenience, we describe $C_k$ just for two different tissue topologies, a tissue organized in stacked hexagonal pillars or a thin layer of similar hexagonal pillars (see Supplementary Methods, §S.1 in electronic supplementary material). In the former case, cells tessellate a three-dimensional space, and we neglect effects at the periphery. In other words, we assume that the number of cells contained within a tissue is larger than the cells at its periphery. Figure 1 illustrates the progression of the number of cells included in subsequent neighbourhoods that can be described analytically as $C_k = 6k^2 + 2$ (electronic supplementary material, equation S1.2 with $s_0 = 6$). For a significantly more constrained topology where only three layers exist $C_1 = 8$ and $C_{k>1} = 6(3k - 2)$ (electronic supplementary material, equations S1.3–4 with $s_0 = 6$). This description permits us to illustrate some

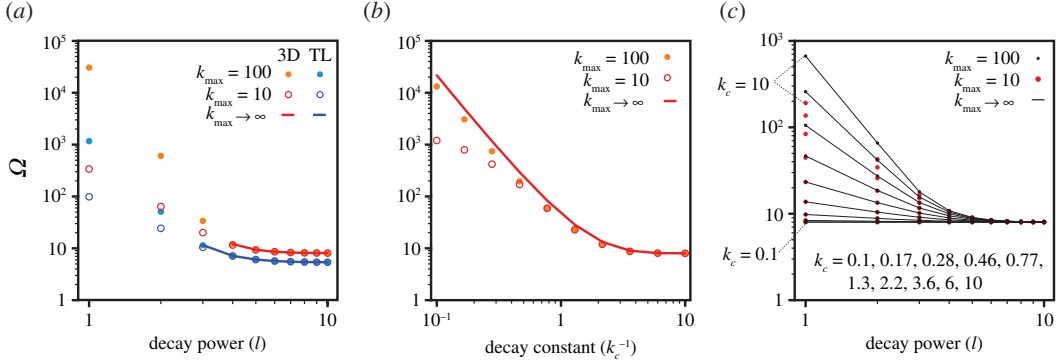

**Figure 2.** Numerical validation of the oncogenic field solutions. (a) Comparison between the numerical evaluation of equation (2.2) for finite values of $k_{max}$ and the estimates of the oncogenic field factor $\Omega$ obtained by the analytical representations for $k_{max} \to \infty$ for a three-dimensional (3D, equation (2.3)) and three-layered (TL, equation (2.4)) tissue model. The solid lines represent the analytical solutions within the limits of its convergence ($l > 2$ for TL in blue and $l > 3$ for 3D in red). The results evaluated over smaller (empty circles) and larger (solid circles) neighbourhood with 10 and 100 cell radii, respectively, represent cases where the assumption of a tissue of infinite extension used to evaluate equations (2.3)–(2.4) is not valid. (b) Identical comparisons as described in (a) but for the exponential decay model for a three-dimensional tissue. Both the analytical solution (equation (2.5), $k_{max} \to \infty$) and numerical estimates of the finite series (equation (2.2), $k_{max} = 10, 100$) converge to the value of $\Omega \sim 8$ for steep decays. (c) Values for $\Omega$ computed for a general case where the oncogenic field decays jointly as the inverse of a power law and exponentially (equation (2.6)). Equation (2.6) (solid lines) is compared with the finite sums ($k_{max} = 10, 100$) for the same parameter sweep shown in (a) and (b), i.e. with the inverse power from 1 to 10 and with a decay constant $k_c$ from 0.1 to 10.

analytical examples about the possible effects of tissue organization on the probability of cooperation between mutations.

## 2.2. Oncogenic field effect

Without loss of generality, we assume that the interaction between two mutant cells is mediated by a shared diffusible product [3], for instance a growth factor or a metabolite. Eldar *et al.* [19] have modelled how the concentration of a signalling molecule (a morphogen) secreted by a cell decays in space. Typically, the morphogen concentration is abated by passive diffusion and linear degradation resulting in exponentially decaying concentration gradients. However, ligand–morphogen interactions can induce nonlinear mechanisms of morphogen degradation resulting in power law decays. Therefore, we first analyse the decay of an oncogenic field akin to morphogen gradients using power or exponential decays because of their physiological relevance [19–22].

For the case of a power function ($p_n(k) = p_{n0}k^{-l}$) and a three-dimensional tissue described by hexagonal pillars ($C_k = 6k^2 + 2$), the factor $\Omega$ can be described analytically (see Supplementary Methods, §S.2 and equation S.2.4 with $s_0 = 6$ in electronic supplementary material for proof) as follows:

$$\Omega(l) = 6\zeta(l-2) + 2\zeta(l), \tag{2.3}$$

where $\zeta$ is the Riemann Zeta function and is finite only for an argument larger than one (here $l > 3$). Therefore, for a large interconnected tissue, oncogenic biochemical gradients induced by a mutant cell must decay very steeply for non-cell-autonomous mechanisms not to dominate. In the limiting case where only the 1-neighbourhood is relevant for transformation ($l \to \infty$), the Riemann Zeta function converges to unity and therefore $\Omega = 8$. This is just the number of cells in direct contact with the reference cell ($C_1$), showing mathematical consistency and providing a lower boundary to $\Omega$ in the case of small effects in a very constrained topology. Conversely, for shallower gradients where the Riemann Zeta function does not converge ($l < 4$), these probabilities will be significantly larger. We obtained these results modelling tissues of non-finite extensions to derive analytical solutions. However, through numerical estimations, it is simple to demonstrate how these observations are generally valid and correct also for small volumes of cells (figure 2a,b). For example, in a small neighbourhood with a radius of 10 cells, $\Omega \sim 11.5$ ($l = 4$) and $\Omega \sim 340$ ($l = 1$), values (figure 2a, solid circles) that reach 12 and $3 \times 10^4$, respectively, for a neighbourhood with a radius of 100 cells (figure 2a, empty circles). Similarly, we demonstrate that for a thin three-layer tissue (see

Supplementary Methods §S.2 and equation S.2.8 in electronic supplementary material for proof),

$$\Omega = 12\zeta(l-1) - 8\zeta(l) + \frac{4}{3}. \tag{2.4}$$

This series converges for $l > 2$, it assumes a value of 11.7 for $l = 3$, and numerical estimations show that $\Omega$ reaches values of approximately 24 and approximately 50 for $l = 2$ within a limiting neighbourhood with a 10 or 100 cell radius, respectively (figure 2a, empty and solid circles). In the limit case where only the first neighbourhood is relevant ($l \to \infty$), $\Omega \sim 5.3$. Therefore, even within this rather constrained topology, $\Omega$ obtains rather large values.

Gradients described by power functions are shallower than exponentially decaying gradients at longer distances. Although both gradients are physiologically relevant, power-like functions might overestimate $\Omega$. It can be readily demonstrated (equation (2.5), see also Supplementary Methods §S.3 in electronic supplementary material) that even for steep gradients decaying of a third at every cell distance ($k_c = 1$), $\Omega$ can assume double-digit values (figure 2b).

$$\Omega = e^{k_c^{-1}} \frac{(2+s_0)e^{2k_c^{-1}} + (4+s_0)e^{k_c^{-1}} + 2}{\left[e^{k_c^{-1}} - 1\right]^3}. \tag{2.5}$$

The analysis of power law and exponential decays are rather instructive, and they are often used to model morphogen gradients as a solution to the reaction–diffusion equation for one-dimensional problems and for specific three-dimensional architectures. We can also demonstrate (see Supplementary Methods §S.4 in electronic supplementary material) that for $p_{nk} = p_{n0}k^{-l}e^{-(k-1)k_c^{-1}}$, i.e. when the oncogenic gradient jointly decays as an inverse power law and exponentially,

$$\Omega = e^{k_c^{-1}}[2Li_l(e^{-k_c^{-1}}) + s_0 Li_{l-2}(e^{-k_c^{-1}})], \tag{2.6}$$

where $Li$ is the polylogarithm. This analytical solution describes the expectation for $\Omega$ for an oncogenic field induced by stochastic (mutationally driven) point-sources of shared resources in an ideal three-dimensional tissue (figure 2c) in the presence of degradation. Once again, at the limit for a fast decaying concentration gradient, the value of $\Omega \sim 8$, long-distance interactions ($k_c \gg 0$) can drastically increase the magnitude of $\Omega$ and with high values found also for small clusters of cells (figure 2c, blue and red circles).

We can thus infer a general consideration from the mathematical description of the proposed case studies that are aimed to exemplify the possible synergy between the mutational process and non-cell-autonomous effects. Unsurprisingly, the specific tissue geometries and the properties of concentration gradients result in rather different magnitudes of an 'oncogenic field'. However, either through *juxtacrine* (contact-dependent) or *paracrine* (short or long distance) signalling, mutations in tissue neighbourhoods that can cooperate through cell-to-cell communication are likely to have a significant role in oncogenesis, in addition to mutations co-occurring within a cell.

## 2.3. Cell-autonomous time horizon

So far, we have discussed *if* and *how likely* mutationally driven non-cell-autonomous mechanisms might be; next, we address the question about *when* these mechanisms are more likely to occur. Indeed, we have shown that non-cell-autonomous mechanisms can increase the probability that mutations contribute to carcinogenesis by a factor $\Omega$. A corollary to this observation is that cooperation between non-transformed cells might contribute to tumour initiation earlier than cell-autonomous mechanisms. For simplicity, we consider only the mutational process and neglect $p_a$, $p_n$ and $p_{n0}$. One cell accrues pairs of mutations at the rate $\rho_0$ but within a neighbourhood cooperating cells accrue mutations at an apparent rate of $\rho_0\sqrt{\Omega}$. First, we tested this simple mathematical inference with Monte Carlo simulations (figure 3 and Methods, §4.2). We simulate the independent and stochastic appearance of four types of mutations (A, B, C and D) at a rate of $\rho_0 = 10^{-6}$ mutations/day on a lattice of $N = 10^6$ cells with 2000 replicates. When one cell accrues mutations A and B, it is flagged as an AB mutant; when a cell becomes a C mutant and in its neighbourhood, there is a D mutant, the C mutant will be listed as a CD (cooperating) clone. The average time for a double-mutant cell to appear ($\langle t_{AB}\rangle$) is $2.43 \pm 0.04$ simulation years (mean ± standard deviation computed over five independent Monte Carlo simulations each made of 2000 runs with 1 year defined as 365 simulation days). The distribution of $\langle t_{AB}\rangle$ values depends only on $\rho_0$ and $N$ but not on $\Omega$ and, therefore, we show the average of the four distributions of $\langle t_{AB}\rangle$ values (figure 3a, black curve) as reference for the cooperating mutants. The distribution of the waiting times

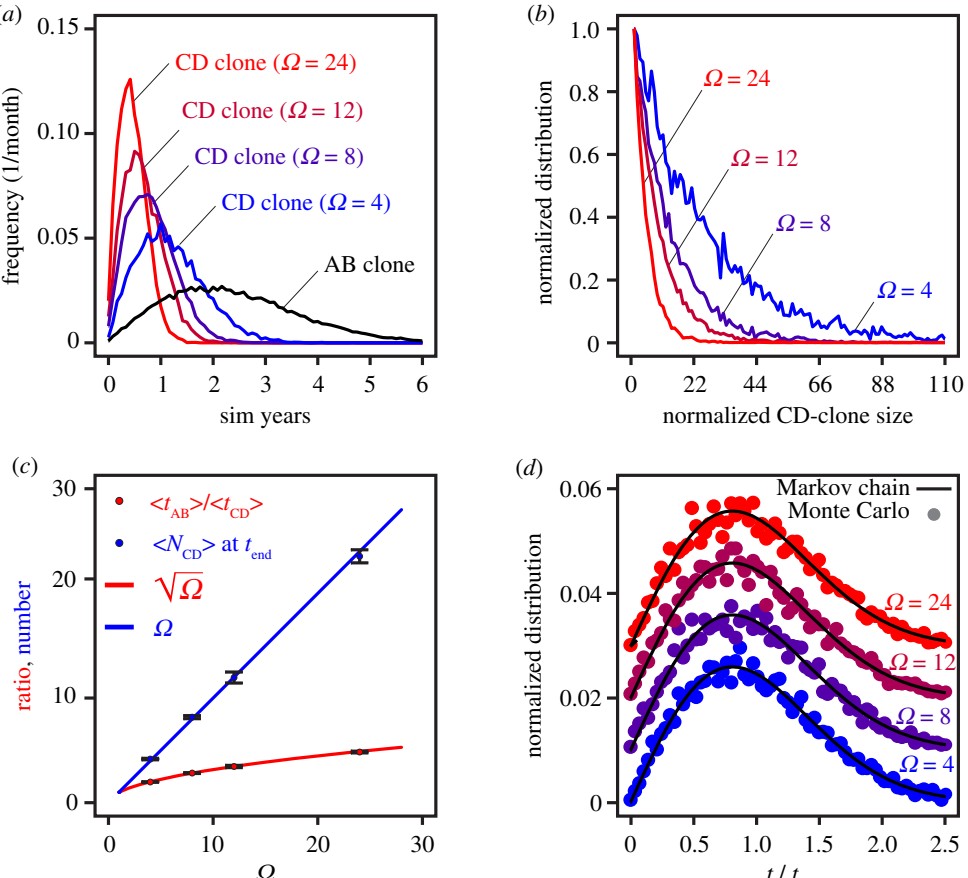

**Figure 3.** Monte Carlo simulations of the cell-autonomous time horizon. (a) Probability distribution of the waiting times for the occurrence of the first two co-occurring mutations (AB clones, black curve) or for the first cooperating mutations (CD clones, coloured curves) through non-cell-autonomous mechanisms for $\Omega$ values equal to 4, 8, 12 and 24. The coloured curves are the average of the five independent simulations runs (see Methods). The black curve is the average of 20 (four values of $\Omega \times 5$ repeats) runs as each simulation had its own AB-control. (b) Distribution (normalized to maximum for better visualization) of the number of CD clones at the end of the simulations ($t = t_{end}$). $t_{end}$ is the time at which at least one AB and one CD clone are detected. (c) The average time ($\langle t_{CD}\rangle$) at which the first cooperating CD clone is observed scales with the square root of $\Omega$ (red line) compared with the average time ($\langle t_{AB}\rangle$) at which the first AB-mutant appears. The average number of cooperating mutations within a neighbourhood at $t = t_{end}$ ($\langle N_{CD}\rangle$) scales as $\Omega$ (blue curve). Errors are standard deviations for five independent sets of Monte Carlo simulations. (d) Distribution of the waiting times for CD clones as shown in (a) but replotted on a new time-base defined, for each curve, as $t/t_{\Omega}$. $t_{\Omega}$ was predicted by the Markov Chain model using equation (2.8b). The distributions predicted by the Markov Chain model (equations (2.7), black curves) fully overlap with the result of the numerical simulations (circles). For better comparison, the distributions were normalized to the sum and offset with a constant.

for the appearance of CD clones depends on the value of $\Omega$ (figure 3a, coloured curve) and exhibits an average time ($\langle t_{CD}\rangle$) of about $1.23 \pm 0.02$, $0.86 \pm 0.03$, $0.70 \pm 0.06$ and $0.50 \pm 0.04$ months for $\Omega$ values equal to 4, 8, 12 and 24, respectively—scaling as $\Omega^{-0.5}$ (figure 3c). Thus, we can define $t_a$ ($\langle t_{AB}\rangle$ in our simulations) as the average time for a tissue of N cells to accrue two mutations, which is inversely proportional to $\rho_0 N$. By definition, $t_a$—the time horizon after which cell-autonomous mechanisms might dominate—is preceded by a latency period during which single mutations are more likely. However, our model predicts the existence of a significantly long period $t_{\Omega} = t_a \Omega^{-0.5} \leq t < t_a$ when mutationally driven cooperation between adjacent cells is more likely than mutationally driven cell-autonomous mechanisms to occur.

For instance, in the limiting case where only the first neighbourhood significantly contributes to tumour initiation ($\Omega = 8$), during approximately 65% of the time interval preceding $t_a$, clonal cooperation is likely to be a fundamental mechanism that synergizes with the mutational process to support partially transformed clones. For $\Omega$ values of 4, 12 and 24, this interval will be about 50%, 70% and 80% of $t_a$. The scaling of $t_{\Omega}$ as $t_a \Omega^{-0.5}$ is shown in figure 3c from the plot of $\langle t_{AB}\rangle/\langle t_{CD}\rangle$ (figure 3c, red) and the scaling of the number of mutations in a given neighbourhood defined by $\Omega$

at $t_a$ is shown from the plot of $\langle N_{CD} \rangle$ (figure 3c, blue). Because of the stochastic nature of the mutational process, the distribution of waiting times for double mutants (AB and CD) are broad. This heterogeneity results in no CD cooperating clones in about 40%, 20%, 15% and 8% of the simulation runs with $\Omega$ 4, 8, 12 and 24, respectively. In the majority of the cases, however, cooperating mutants preceded AB clones and for larger values of $\Omega$ the probability for mutations to appear within a tissue neighbourhood is so high that CD mutants are likely to reoccur multiple times randomly (figure 3b).

Next, to better explain and generalize the origin of the scaling factor $\Omega^{-0.5}$, we modelled the mutational process as a discrete-time Markov chain [16,17] (see Supplementary Methods §S.5 and Mathematica Notebooks in electronic supplementary material). We show that for a 'two-hits' model, the distributions shown in figure 3a can be described analytically as follows:

$$p_{AB}(t) \approx 2tN\rho_0^2 e^{-t^2 N\rho_0^2} \tag{2.7a}$$

and

$$p_{CD}(t) \approx 2tN\rho_0^2 \Omega e^{-t^2 N\rho_0^2 \Omega}. \tag{2.7b}$$

Figure 3d shows the very good match between the Monte Carlo simulations and the Markov chain model. Using equations (2.7), we can then estimate the cell-autonomous time horizon as the average time required to observe the first AB clone in N cells (equation (2.8a)) and, similarly, the average latency to observe the first cooperating CD clone (equation (2.8b)).

$$t_a = \langle t_{AB} \rangle \cong \frac{1}{2\rho_0} \sqrt{\frac{\pi}{N}} \tag{2.8a}$$

and

$$t_\Omega = \langle t_{CD} \rangle \cong \frac{1}{2\rho_0} \sqrt{\frac{\pi}{N\Omega}} = t_a \Omega^{-0.5}, \tag{2.8b}$$

where $t_\Omega$ is indeed rescaled by a factor $\Omega^{-0.5}$ relative to $t_a$ as observed in the Monte Carlo simulations. Equations (2.8) provide estimates for $t_a = 2.43$ years and $t_\Omega = 1.21$, 0.86, 0.70 and 0.50 years (for $\Omega = 4$, 8, 12 and 24, respectively), values that are in excellent agreement with the numerical simulations. We note that for the more general case where $m$ mutations cooperate through non-cell-autonomous mechanisms, the scaling factor $\Omega^{-0.5}$ would assume the form $\Omega^{-d/m}$ shown in electronic supplementary material, equation S.5.9, with $d$ representing the number of mutations that cooperate at distance defined by the parameter $\Omega$. For example, in the case where a mutant cell C interacts with a mutant cell D via paracrine effects and D reciprocates, the scaling factor is $\Omega$ ($d = m = 2$). However, if a C-mutant with $m - 1$ mutations benefits from a D-mutant (with a single mutation) in its neighbourhood, this scaling factor is $\Omega^{-1/m}$.

We note that even assuming a role for synergy between the mutational process and cell-to-cell communications during the earliest steps in oncogenesis ($t < t_a$), chance will determine the first occurrence of co- or cooperating mutations (as a function of $\Omega$), possibly influencing the evolutionary trajectory of a tumour and contributing to tumour heterogeneity.

## 3. Discussion

A role for non-cell-autonomous mechanisms in cancer is well established, often as a mechanism of interaction between cancer cells and the surrounding tissue [2,23–26]. The cooperation of non- or partially transformed clones as a driving force underlying oncogenesis has also been hypothesized [3], and there is nowadays accumulating evidence suggesting that a description of oncogenesis focused exclusively on cell-autonomous mechanisms might under-represent the importance of oncogenic signalling in cancer [5,9,26].

Experiments in *Drosophila melanogaster* have also shown that inter-clonal cooperation between mutants harbouring an oncogenic KRAS mutation or inactivation of the tumour suppressor *scrib* can support tumorigenesis mediated by JNK and JAK/STAT signalling [27]. Recently, Marusyk *et al.* [9] have used a mouse xenograft model to test the effects of clonal heterogeneity, demonstrating that clones expressing the chemokines IL11 are capable of stimulating overall tumour growth through a non-cell-autonomous mechanism, while clonal interference maintains genetic intra-tumour

heterogeneity [9]. Similarly, Inda *et al.* [5] have shown how intra-tumour heterogeneity observed in glioblastoma can be maintained through cross-talk between mutants harbouring a ΔEGFR that secrete IL6 and LIF to support fitness in clones with EGFR amplification [5]. Cleary *et al.* [28] has also shown that WNT-producing HRAS wild-type clones can support tumorigenicity and clonal heterogeneity by cooperating with clones harbouring mutant oncogenic HRAS [28]. These observations support the emerging notion that intra-tumoural heterogeneity is often of poly-clonal origin and is an active process supported by non-cell-autonomous mechanisms. However, the role for poly-clonality and clonal cooperation during earlier stages of oncogenesis can be seen in contradiction with the low estimates of mutational rates in cancer [10]. Furthermore, it is unclear if clonal cooperation has a role during early oncogenesis or only at later stages when a heterogeneous tumour is established [28].

Aiming to contribute to filling this gap in knowledge, we developed simple analytical and computational models for mutation-driven oncogenesis in the presence of cell-to-cell communication. Using similar assumptions used to model mutationally driven oncogenesis, we have studied *if*, *how* and *when* it is likely that cell-to-cell communication might cooperate with the mutational process from the perspective of basic principles. Our analysis raises provoking observations on the earliest steps in oncogenesis. We show that, irrespective of the background mutational rate, if a set of transforming mutations are sufficiently likely to occur within a single cell in the lifetime of a patient, an equally rare yet oncogenic set of mutations are equally (or more) likely to contribute to tumorigenesis through non-cell-autonomous mechanisms. We have introduced the parameter $\Omega$, which captures the impact of tissue organization and non-cell-autonomous mechanisms on cancer evolution. We modelled non-cell-autonomous mechanisms in analogy to morphogens during embryonic developments. $\Omega$ describes the magnitude with which paracrine, juxtacrine and other mechanisms mediated by shared substrates (e.g. growth factors and metabolites) might impact the transformation of a cell or clone. As such, $\Omega$ represents an oncogenic field effect, where oncogenic fields have the opposite outcome of morphogens by contributing to the de-regulation of tissue homeostasis. Furthermore, we have identified a stage of oncogenesis during which clonal cooperation might not simply coexist with the clonal competition but even dominate before the emergence of clones capable of growing autonomously. With the help of our model, experimentally, the problem is reduced to the measurement of quantities such as $p_n0$ and $p_a$ or the abundance of genes that, once mutated, can drive oncogenesis by non-cell-autonomous mechanisms. We argue that the magnitude of the oncogenic field effect ($\Omega$) and the prediction of an autonomous time horizon suggest a significant role for mutationally driven and non-cell-autonomous mediated poly-clonal evolution of cancer during, at least, a very early stage of oncogenesis.

The model described here is purposely simple aiming to illustrate the basic principles emerging by the cooperation of the mutational process with non-cell-autonomous mechanisms [3], a phenomenon that, to our knowledge, is often neglected when models of somatic evolution of cancer are studied analytically [11]. For this reason, we did not include the description of more complex and important features of real tissues such as clonal dynamics, tissue homeostasis, tissue mechanics and other mechanisms for gradient formation of biomolecules. Each of these processes can change considerably the magnitude of the effects we described. The concentration gradients on their own, for instance, can be enhanced by compartmentalization, abrogated by diffusion into lumens or the vascular system, or affected by systemic alterations of shared resources (e.g. hormones, lipids). If a proliferative tissue is considered, with a fitness advantage for cooperative clones compared with wild-type cells, the presence of these non- or partially transformed clones could be even more significant, increasing the probability to accrue further mutations at a faster pace and shaping the initial period of oncogenesis.

However, tissues are complex systems and diverse mechanisms of tissue homeostasis in different tissues might conflict with this perspective. In the case of a fast self-renewing tissue like the intestinal epithelium within which the cell-of-origin for common tumours is likely to be a stem cell [29], the highly compartmentalized stem-cell niche might pose an effective barrier to oncogenic field effects. The intestinal epithelium is one of the most proliferative tissues subject to a high mutagenic burden and it has been broadly studied both mathematically [13,30] and experimentally [29,31,32]. A small group of adult stem cells reside within the colonic crypt and maintain the homeostasis of the villi lining the intestine [29]. Within the crypt and the villus, the balance between proliferation and differentiation is maintained by a complex network of signals (e.g. WNT, Notch, BMP and EGF) generated by specialized Paneth cells within the crypt and cells within mesenchyme lining the crypt [32]. Mutations in the WNT (e.g. APC or CTNNB), EGFR (e.g. KRAS, PIK3C or BRAF) and TGF-β

(e.g. SMAD4) signalling pathways gradually render cells independent from niche signals to grow autonomously and promoting cancer. While it is still more likely that two mutations are acquired within adjacent stem cells rather than within one cell, neutral genetic drift or selection fix or purge genetic mutations within the crypt that will be thus, most of the times, monoclonal [29]. However, not all mutations occur in the stem cells within the crypt [33] and tissue homeostasis is likely to be maintained by crypt fusion and fission leading to field cancerization [15] supporting the possibility that partially or non-transformed clones might interact not within a monoclonal crypt but between patches harbouring different mutations [29,34].

These considerations might hold true also for other highly proliferative tissues, such as the well-described human epidermis and the oesophageal epithelium [35–38]. In these tissues, homeostasis is maintained by a balance between the probability for progenitor cells to divide symmetrically or asymmetrically, giving birth to two progenitor cells, two differentiating cells, or—more commonly—one differentiating and one progenitor cell [35,38]. Alcolea *et al.* [35], for example, have shown that mutations in the Notch pathway reduce the probability for a progenitor cell to generate two differentiating cells and induce wild-type cells to differentiate. In combination with P53 mutations, this cell fate imbalance leads to field cancerization [35]. Considering the experimental observations on later stages of carcinogenesis we have already discussed [5,6,8,9,27,28] and our results, it is conceivable that mutations might also cause cell fate imbalance through non-cell-autonomous mechanisms during early oncogenesis. As cell fate determination and the occurrence of mutations are stochastic processes, the role of non-cell-autonomous mechanisms not only might vary across different tissues depending on their organization but also within the same tissue of origin.

We described in figure 3 that the occurrence of a double-mutant clone that might acquire a fitness advantage autonomously, even if less likely, can still precede the occurrence of two cooperating single-mutants. Similarly, genetic drift and selective pressure could either maintain or collapse one or both of the cooperating populations [9]. Therefore, each tissue and each tumour might be affected differently by non-cell-autonomous mechanisms, mechanisms that could alter the evolutionary trajectory of tumours that later acquire full independence from cooperating clones, thus also contributing to tumour heterogeneity [7,9]. As computational modelling of multicellular tissues can describe complex homotypic and heterotypic interactions, including short- and long-range interactions and tissue mechanics [14,39,40], computational models rather than analytical tools might be more appropriate to investigate the possible role for 'oncogenic fields' in complex mutagenic environments.

The somatic mutation theory is the prevailing model of carcinogenesis which has been described mathematically with several different approaches [11,13,18,41–44]. Modelling work based on evolutionary game theory (e.g. [2,45,46]) and analysis of clonal heterogeneity [9], among others, have already highlighted the importance of clonal competition and cooperation in cancer. However, mathematical models of somatic mutation theory often do not include cooperation between non- or partially transformed mutants, particularly when studying the earliest stages of carcinogenesis. Through the lens of the 'toy model' we presented here, we show that tissue organization and cell-to-cell communication might cooperate synergistically with a mutationally driven process, particularly during the early stages of carcinogenesis. We emphasize that our work is not in contradiction with prevailing models of oncogenesis, as it is based on similar assumptions but includes explicitly the possibility that non-transformed mutant cells can cooperate. The mathematical analysis we presented was not elaborated to capture more complex phenomena occurring during oncogenesis. However, our analysis suggests that to improve our understanding of carcinogenesis, the identification of the genes and the shared resources that can mediate clonal cooperation—such as growth factors (e.g. mitogens, interleukins, *etc.*) or metabolic by-products that are often at the basis of cooperative behaviour in lower organisms [47–49]—might be of fundamental importance.

# 4. Methods

## 4.1. Analytical methods

The detailed derivation of equations (2.1)–(2.8) shown in this work is provided in electronic supplementary material, Methods. The discrete-time Markov chain model is described in electronic supplementary material, §S.5 and its implementation is also provided in an annotated Mathematica

notebook '*firstpassageproblem_v2.nb*' included in this submission and available also from the GitHub repository *alesposito/CloE-PE* [50].

## 4.2. Numerical simulations

The numerical evaluation of the analytical results (figure 2*a–c*) was performed with the Matlab script '*analytical_and_numerical_comparisons.m*' (v4, Mathworks, version 2018) freely available from the GitHub repository *alesposito/CloE-PE* [50]. The numerical estimations simply compare the value obtained from the approximated analytical solutions described in the appendices to direct numerical estimate computed on given neighbourhoods with features described in the main text. For the case of a three-dimensional tissue (§2.2 and Supplementary Methods §S.2 in electronic supplementary material), the values of equation (2.3) (electronic supplementary material, equation S.2.4 in its more general form) were compared with those of the finite series $\Omega = \sum_{k=1}^{kn} (k^2 s_0 + 2)k^{-l}$ for different power functions. Data shown in figure 2*a* are computed with $kn$ equal to 50 and 100, $l$ ranging from 1 to 10, and $s_0 = 6$. Within the same parameter space, we compare the analytical description of a three-layer thin tissue represented by equation (2.4) (or electronic supplementary material, equation S.2.8) to the numerical estimates of $\Omega = 2 + \sum_{k=1}^{kn} s_0(3k - 2)k^{-l}$. Similarly, figure 2*b* shows a comparison between electronic supplementary material, equation S.3.6 describing $\Omega$ for a three-dimensional tissue with an oncogenic field decaying as an exponential function (Supplementary Methods §S.3 in electronic supplementary material) and the finite series $\Omega = \sum_{k=1}^{kn} (k^2 s_0 + 2)e^{-(k-1)k_c^{-1}}$. Also in this case, the parameters used in the numerical evaluations were $kn = 50$ and 100, and $s_0 = 6$ with the inverse of the decay constant $k_c^{-1}$ spanning the 0.1 to 10 range. Last, for the distribution jointly decaying as a power and exponential function (Supplementary Methods §S.4 in electronic supplementary material, and figure 2*c*), numerical estimates of $\Omega = \sum_{k=1}^{kn} (k^2 s_0 + 2) \, k^{-l} e^{-(k-1)k_c^{-1}}$ were compared with equation (2.5) (or electronic supplementary material, equations S4.3 and S.4.4 for $l = 1$) on the same parameter space described for the other cases.

The Monte Carlo simulations (figure 3) used to evaluate the relationship between the time horizon for cell-autonomous mechanisms ($t_a$) and non-cell-autonomous mechanisms are available as the Matlab script '*polyclonal_mutation_cooccurence_check.m*' (v7) freely available from the GitHub repository *alesposito/CloE-PE* [50]. We simulated a lattice of $10^6$ cells with a mutational rate equal to $10^{-6}$ mutations per cell per simulated day (simday). At each simday and at each node of the lattice, four random numbers ($n_m$, with $m$ = A, B, C or D) were drawn from uniformly distributed numbers in the [0,1] interval. For any of the indexes A, B, C or D where $n_m$ was lower than or equal to $10^{-6}$, the correspondent cell was switched from non-mutant to mutant. Cells were then allowed to accumulate these four mutations for a maximum of 100 000 simdays. When a cell acquires both A and B mutations, an AB-mutant cell is established and logged. When a D mutation appears in a neighbourhood of a C-mutant, a CD-cooperative clone is logged. CD-cooperative mutants are detected using convolution filters that detect the co-occurrence of a D-mutant within the centre of a reference neighbourhood and C-mutant in its immediate vicinity. The publicly available code implements the following neighbourhoods scans equivalent to $\Omega$ value of 4, 8 and 12. For $\Omega = 4$, detection in position north (N), east (E), south (S) and west (W); for $\Omega = 8$, as for the previous case but with the addition of NE, NW, SE and SW; for $\Omega = 12$, as for the previous case but with the addition of one non-adjacent cell in N, E, S and W position. As soon as at least one AB and one CD-cooperating clone occurs, the simulation is interrupted. Simulations are then repeated 2000 times and the distributions of the appearance of first AB or CD clones, and number of CD clones at the appearance of an AB clone are generated. When results are presented in simulation years, the number of simulated days was simply divided by 365.

The scripts were run on a Dell Precision 5810 workstation using an Intel Xeon E5-1625 CPU and 64 GB RAM. The 20 Monte Carlo simulations shown in figure 3 (five repeats of the four conditions) are computationally intensive and were run in parallel for about 9 days.

The code and data used to generate figure 3 are also available from the GitHub repository *alesposito/ CloE-PE* [50] (*figure3.zip*).

Data accessibility. All the code and data used in this paper are available at the GitHub repository alesposito/CloE-PE (https://doi.org/10.5281/zenodo.4410222).

Competing interests. We declare we have no competing interests.

Funding. A.E. acknowledges the financial support provided by the CRUK with a multi-disciplinary project award (OncoLive, C54674/A27487), pump-priming funds from the CRUK Cambridge Center (C9685/A25117, C9685/A28397). A.E. also acknowledges financial support from Medical Research Council program grants (MC_UU_12022/1 and MC_UU_12022/8) awarded to Prof. Ashok Venkitaraman.

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
