## [Peer Review File · Royal Society Open Science]

Review History

RSOS-201532.R0 (Original submission)

Review form: Reviewer 1

Is the manuscript scientifically sound in its present form?

Yes

Are the interpretations and conclusions justified by the results?

Yes

Is the language acceptable?

Yes

Do you have any ethical concerns with this paper?

No

Have you any concerns about statistical analyses in this paper?

No

Recommendation?

Accept with minor revision (please list in comments)

Comments to the Author(s)

The author has addressed the vast majority of points that I raised in my review of the original manuscript. However, although they have added additional references related to existing modelling work addressing the interactions of mutations in clones, they have not made it clear how their work sits with regard to these models. I would therefore like to see additional paragraphs added to the Introduction/Background sections to address this important point, clarifying the context of their modelling approach and the novelty/further insight that it provides.

Regarding figure 3 - the grey lines are almost invisible, so I would suggest further revision of this figure to improve visual clarity.

Review form: Reviewer 2**Is the manuscript scientifically sound in its present form?**

Yes

Are the interpretations and conclusions justified by the results?

Yes

Is the language acceptable?

Yes

Do you have any ethical concerns with this paper?

No

Have you any concerns about statistical analyses in this paper?

No

Recommendation?

Major revision is needed (please make suggestions in comments)

Comments to the Author(s)

The manuscript by Esposito examines the role played by cell-cell communication in the early stages of oncogenesis. The postulate is that it is more likely for deleterious mutations to accumulate in a clonal neighborhood than in a single cell, and therefore this route may be relevant for oncogenesis. In itself this suggestion is not surprising since the notion that signaling pathways and inter-cellular communications are relevant to oncogenesis is central at least to some viewpoints of the origins of cancer in somatic cells and tissues. The proposed mechanism is supported by estimates of the size of the relevant neighborhood ($\Omega \approx 10-100$ cells), and Monte-Carlo simulations of a simple model which demonstrates that with all other things being equal, the formation of mutant clones is faster by a factor $\Omega^{1/2}$ (fig 3) compared to the autonomous accumulation of similar mutations in a single cell. Whilst I think the actual importance of clonal co-operation versus cell autonomy must be evaluated on a case-by-case basis, I do accept the argument by the author that the models presented here make the point in quite a clear and dramatic manner. Therefore I support publication on the grounds of general interest, with revision to take into account the comments below.

The material seems a little disjointed. I recognize there is a desire to partition the presentation into ‘materials and methods’ and ‘results’, but in this case it seems to introduce unhelpful and artificial barriers to the natural progression of the argument. It might be better to collect together all the material for the Ω estimates and present it first, as a complete whole, before moving on to presenting the Monte-Carlo simulations.

In general, I found the presentation a bit labored and over-elaborated in places. For example the first two equations in section 3.2 could easily be combined (the sum is trivial). Likewise the presentation in sections 3.3 and 3.4 could be greatly condensed.

In fig 2 and the accompanying discussion in section 3.6, I think this is not really a comparison between ‘analytic’ and ‘numerical’ results as such, but rather an assessment of the effect of truncating the sums. I don’t think anyone doubts the ability of MATLAB or another numerical platform to evaluate the truncated sums, so the benchmark ‘analytic’ results would be better stated as results for $k_{\max} \rightarrow \infty$ (fig 2 annotation), in my opinion.

In section 4.1 there is a statement “The formalism for exponentially decaying oncogenic fields is less elegant”. I presume the author means the expressions for pure power-law decays are more succinct? However, there is a much better biophysical motivation for exponential decay (it results from the competition between diffusion and degradation), or for exponentially-truncated power-laws, so I don’t see how elegance (or succinctness) can be a relevant criterion here (cf “elegance is for tailors”, attributed to Boltzmann).

Please provide error estimates if available for the mean first passage times quoted in section 4.2 (ie, 29 months for first appearance of double mutant cells; and 14.7, 10.1, 8.5 and 6.2 months respectively for $\Omega = 4, 8, 12$ and 24, for the appearance of mutant clones). Also, for clarity, please define ‘months’ (and ‘years’) in terms of the number of days (see also below).

I did not understand the normalisation being used in fig 3a. If the vertical axis is frequency ‘per year’, then I think these curves are all too low. For example the support for the $\Omega = 24$ curve is almost all concentrated in the first year, so wouldn’t one expect the peak to be $> O(1)$ in units of years^{-1} (the integrated area under the curve should be unity, for a probability distribution). Also, when I try to compare the curves with the analytic results derived below, they all seem too low by a constant factor.

Please number all displayed equations! This is Fisher’s rule - see “What's Wrong with these Equations?”, N. D. Mermin, *Physics Today* 42, 9 (1989) [a PDF can be found by searching for the title].

[Also there are no page numbers in the manuscript, which makes it harder to refer to the text.]

The main remaining problem I have with the manuscript as it stands is that there is no proper discussion that I could see of the possible origin of the $\Omega^{1/2}$ scaling result, nor is there any attempt to explain the apparently exact scaling laws in fig 3b. But I note that in the Monte-Carlo model the appearance of AB mutant cells or CD mutant clones is essentially a first passage time problem [see for example N. van Kampen, *Stochastic Processes in Physics and Chemistry* (North-Holland, 1981)]. As such, I think the results are fairly easy to rationalize with a modicum of statistical insight. I first give a heuristic analysis and then present what I think is likely to be an exact solution to the AB mutant cell problem, and a good approximation for the CD mutant clone case. From these one can make some predictions which could be tested. The inclusion of these tests, supported by some version of the arguments made below, would I think strengthen the manuscript.

Given that the mutations occur with probability $p \ll 1$ per day, after k days (with kp small) the probability that a cell will acquire an A mutation is $\approx kp$, and the probability that it will acquire a B mutation is likewise. Therefore in a population of size N , the expected number of AB mutants will be $N_{AB} \approx Nk^2p^2$. Likewise the probability that a cell will acquire a C mutation is $\approx kp$, and the probability that a cell in its neighborhood of size Ω will acquire a D mutation is $\approx \Omega kp$, and therefore the expected number of CD mutant clones will $N_{CD} \approx N\Omega k^2p^2$. From these one can estimate the mean first passage times as the number of days for one AB mutant cell or one CD mutant clone to appear; these being respectively $\langle t_{AB} \rangle \approx 1 / p\sqrt{N}$ and $\langle t_{CD} \rangle \approx 1 / p\sqrt{(\Omega N)}$. For example with $p = 10^{-6}$ and $N = 10^6$, one has $\langle t_{AB} \rangle \approx 1000$ days ≈ 30 months, which close to the reported value (but see below). Similarly one finds $\langle t_{AB} \rangle / \langle t_{CD} \rangle \approx \sqrt{\Omega}$, and that the expected number of CD clones when one AB mutant appears is given by $N_{CD} = N\Omega k^2p^2$ when $N_{AB} = Nk^2p^2 = 1$, and is therefore $N_{CD} = \Omega$. These two predictions exactly fit the scaling laws in fig 3b.

But I think the AB mutation problem can be solved exactly since the accumulation of mutations in each one of the N cells can be viewed as an independent process. For each cell, the probability of acquiring both mutations can be represented by a Markov chain with a state space $\{W, A, B, AB\}$, with transitions $W \rightarrow A$, $W \rightarrow B$, $A \rightarrow AB$, and $B \rightarrow AB$ all occurring with probabilities p (for simplicity I omit the direct transition $W \rightarrow AB$ which occurs with probability p^2 , but its inclusion would be straightforward). The transition matrix for such a Markov chain is $\{\{1 - 2p, p, p, 0\}, \{0, 1 - p, 0, p\}, \{0, 0, 1 - p, p\}, \{0, 0, 0, 1\}\}$ (the AB state is adsorbing). The cell starts in the W state (wild-type). Then one can compute exactly the probability that the cell is in the AB state after k days, viz. $\text{Prob}(AB; k) = 1 + (1 - 2p)^k - 2(1 - p)^k$ (for example this can be conveniently done using Mathematica's DiscreteMarkovProcess function). If $p \ll 1$, then $\text{Prob}(AB; k) \approx k(k - 1)p^2$, which is of course to be expected since it is the probability of acquiring the A or B mutation in k days, multiplied by the probability of acquiring the other mutation in the remaining $k - 1$ days. To proceed to the first passage time problem in the ensemble of cells, I note that the probability of a cell NOT being in the AB state after k days is $1 - \text{Prob}(AB; k)$; and consequently the probability of there being no AB mutants in the ensemble of N cells after k days is $[1 - \text{Prob}(AB; k)]^N$. Then, the probability of an AB mutant appearing in the ensemble after k days is $P = 1 - [1 - \text{Prob}(AB; k)]^N$. As far as I can see this is the cumulative distribution function that corresponds to the first passage time problem; and so the problem is solved.

Further progress can be made by taking $\text{Prob}(AB; k) \approx k^2p^2$ which is valid if $k \gg 1$ (but not too large). Then the probability of an AB mutant appearing after k days is $P \approx 1 - (1 - k^2p^2)^N \approx 1 - \exp(-Nk^2p^2)$. With these assumptions, the first passage time distribution itself is $p(k) = \partial P / \partial k = 2Nkp^2 \exp(-Nk^2p^2)$. The mean first passage time is then $k_m = \int_0^\infty dk k p(k) = \sqrt{\pi} / 2p\sqrt{N}$ (which supports the heuristic estimate above), and in terms of k_m one can write $p(k) = (\pi k / 2k_m^2) \exp(-\pi k^2 / 4k_m^2)$. Putting numbers in, and assuming one month is $365 / 12 = 30.4$ days (see above), the mean first passage time $k_m = 886$ days = 29.1 months. Assuming I chose the right number of days in a month, this is in perfect agreement with the result quoted by the author. Likewise, if I plot the first passage time distribution on top of the AB curve in fig 3a, there appears to be good agreement (modulo normalisation; see above).

An analogous calculation can be made for the CD mutant clones if one neglects the possibility that the Ω -domains overlap. The transition matrix in the CD case would be $\{\{1 - 2p, p, p, 0\}, \{0, 1 - \Omega p, 0, \Omega p\}, \{0, 0, 1 - \Omega p, \Omega p\}, \{0, 0, 0, 1\}\}$. The math goes through as above, with the result that the mean first passage time for the appearance of a CD clone is $k_m = \sqrt{\pi} / 2p\sqrt{(\Omega N)}$ (in agreement with the heuristic argument). Written in terms of this value for k_m , the first passage time distribution is the same as for the AB mutant case.

Some predictions can be made from the above:

First, I would expect a scaling collapse for *all* the first passage time distributions shown in fig 3a if the time scale is normalised by the mean first passage time (ie plotted as a function of k / k_m in the above notation). This could easily be tested.

Second, if one takes the cumulative distribution function P , being the probability of an AB mutant (or CD clone) appearing by k days, a plot of $\ln(1 - P)$ against k^2 should be a straight line with slope $-Np^2$ (or $-\Omega Np^2$). This too is easily testable.

Third, the argument indicates that geometry is unimportant, so that the same scaling laws should hold for instance in a 1d linear array of cells. One might test this, but I would not necessarily expect this for the present manuscript.

Finally, from the above analysis it appears that the origin of the $\Omega^{1/2}$ scaling lies in the fact that two mutations are required to form a CD clone, but only the second of these benefits from the 'clonal amplification' effect. In general, if 'n' out of 'm' mutations are 'clonally amplified', then I would expect the scaling to be $\Omega^{n/m}$.

Decision letter (RSOS-201532.R0)

Dear Dr Esposito

The Editors assigned to your paper RSOS-201532 "Cooperation of partially-transformed clones: an invisible force behind the early stages of carcinogenesis" have now received comments from reviewers and would like you to revise the paper in accordance with the reviewer comments and any comments from the Editors. Please note this decision does not guarantee eventual acceptance.

Please submit your revised manuscript and required files (see below) no later than 21 days from today's (ie 04-Nov-2020) date. Note: the ScholarOne system will 'lock' if submission of the revision is attempted 21 or more days after the deadline. If you do not think you will be able to meet this deadline please contact the editorial office immediately.

Please note article processing charges apply to papers accepted for publication in Royal Society Open Science (<https://royalsocietypublishing.org/rsos/charges>). Charges will also apply to papers transferred to the journal from other Royal Society Publishing journals, as well as papers submitted as part of our collaboration with the Royal Society of Chemistry

(<https://royalsocietypublishing.org/rsos/chemistry>). Fee waivers are available but must be requested when you submit your revision (<https://royalsocietypublishing.org/rsos/waivers>).

on behalf of Catrin Pritchard (Subject Editor)
openscience@royalsociety.org

Subject Editor Comments to Author (Catrin Pritchard):

Comments to the Author:

Please could the authors address all of the issues raised by both reviewers and we will then reassess the manuscript. I am a little concerned that the second reviewer has recommended extensive changes and I would look to see all of these being addressed in a revised version.

Associate Editor Comments to Author:

Comments to the Author:

While reviewer 1 is largely satisfied with the changes you have made after transfer from JRSI, the second reviewer has extensive queries that we'd like you to address in a revision, please.

Reviewer comments to Author:

Reviewer: 1

Comments to the Author(s)

The author has addressed the vast majority of points that I raised in my review of the original manuscript. However, although they have added additional references related to existing modelling work addressing the interactions of mutations in clones, they have not made it clear how their work sits with regard to these models. I would therefore like to see additional paragraphs added to the Introduction/Background sections to address this important point, clarifying the context of their modelling approach and the novelty/further insight that it provides.

Regarding figure 3 - the grey lines are almost invisible, so I would suggest further revision of this figure to improve visual clarity.

Reviewer: 2

Comments to the Author(s)

The manuscript by Esposito examines the role played by cell-cell communication in the early stages of oncogenesis. The postulate is that it is more likely for deleterious mutations to accumulate in a clonal neighborhood than in a single cell, and therefore this route may be relevant for oncogenesis. In itself this suggestion is not surprising since the notion that signaling pathways and inter-cellular communications are relevant to oncogenesis is central at least to some viewpoints of the origins of cancer in somatic cells and tissues. The proposed mechanism is supported by estimates of the size of the relevant neighborhood ($\Omega \approx 10$ -100 cells), and Monte-

Carlo simulations of a simple model which demonstrates that with all other things being equal, the formation of mutant clones is faster by a factor $\Omega^{1/2}$ (fig 3) compared to the autonomous accumulation of similar mutations in a single cell. Whilst I think the actual importance of clonal co-operation versus cell autonomy must be evaluated on a case-by-case basis, I do accept the argument by the author that the models presented here make the point in quite a clear and dramatic manner. Therefore I support publication on the grounds of general interest, with revision to take into account the comments below.

The material seems a little disjointed. I recognize there is a desire to partition the presentation into 'materials and methods' and 'results', but in this case it seems to introduce unhelpful and artificial barriers to the natural progression of the argument. It might be better to collect together all the material for the Ω estimates and present it first, as a complete whole, before moving on to presenting the Monte-Carlo simulations.

In general, I found the presentation a bit labored and over-elaborated in places. For example the first two equations in section 3.2 could easily be combined (the sum is trivial). Likewise the presentation in sections 3.3 and 3.4 could be greatly condensed.

In fig 2 and the accompanying discussion in section 3.6, I think this is not really a comparison between 'analytic' and 'numerical' results as such, but rather an assessment of the effect of truncating the sums. I don't think anyone doubts the ability of MATLAB or another numerical platform to evaluate the truncated sums, so the benchmark 'analytic' results would be better stated as results for $k_{\max} \rightarrow \infty$ (fig 2 annotation), in my opinion.

In section 4.1 there is a statement "The formalism for exponentially decaying oncogenic fields is less elegant". I presume the author means the expressions for pure power-law decays are more succinct? However, there is a much better biophysical motivation for exponential decay (it results from the competition between diffusion and degradation), or for exponentially-truncated power-laws, so I don't see how elegance (or succinctness) can be a relevant criterion here (cf "elegance is for tailors", attributed to Boltzmann).

Please provide error estimates if available for the mean first passage times quoted in section 4.2 (ie, 29 months for first appearance of double mutant cells; and 14.7, 10.1, 8.5 and 6.2 months respectively for $\Omega = 4, 8, 12$ and 24, for the appearance of mutant clones). Also, for clarity, please define 'months' (and 'years') in terms of the number of days (see also below).

I did not understand the normalisation being used in fig 3a. If the vertical axis is frequency 'per year', then I think these curves are all too low. For example the support for the $\Omega = 24$ curve is almost all concentrated in the first year, so wouldn't one expect the peak to be $> O(1)$ in units of years^{-1} (the integrated area under the curve should be unity, for a probability distribution). Also, when I try to compare the curves with the analytic results derived below, they all seem too low by a constant factor.

Please number all displayed equations! This is Fisher's rule - see "What's Wrong with these Equations?", N. D. Mermin, Physics Today 42, 9 (1989) [a PDF can be found by searching for the title].

[Also there are no page numbers in the manuscript, which makes it harder to refer to the text.]

The main remaining problem I have with the manuscript as it stands is that there is no proper discussion that I could see of the possible origin of the $\Omega^{1/2}$ scaling result, nor is there any attempt to explain the apparently exact scaling laws in fig 3b. But I note that in the Monte-Carlo model the appearance of AB mutant cells or CD mutant clones is essentially a first passage time problem [see for example N. van Kampen, Stochastic Processes in Physics and Chemistry (North-Holland, 1981)]. As such, I think the results are fairly easy to rationalize with a modicum of statistical insight. I first give a heuristic analysis and then present what I think is likely to be an exact solution to the AB mutant cell problem, and a good approximation for the CD mutant clone case. From these one can make some predictions which could be tested. The inclusion of these tests, supported by some version of the arguments made below, would I think strengthen the manuscript.

Given that the mutations occur with probability $p \ll 1$ per day, after k days (with kp small) the probability that a cell will acquire an A mutation is $\approx kp$, and the probability that it will acquire a B mutation is likewise. Therefore in a population of size N , the expected number of AB mutants will be $N_{AB} \approx Nk^2p^2$. Likewise the probability that a cell will acquire a C mutation is $\approx kp$, and the probability that a cell in its neighborhood of size Ω will acquire a D mutation is $\approx \Omega kp$, and therefore the expected number of CD mutant clones will $N_{CD} \approx N\Omega k^2p^2$. From these one can estimate the mean first passage times as the number of days for one AB mutant cell or one CD mutant clone to appear; these being respectively $\langle t_{AB} \rangle \approx 1 / p\sqrt{N}$ and $\langle t_{CD} \rangle \approx 1 / p\sqrt{\Omega N}$. For example with $p = 10^{-6}$ and $N = 10^6$, one has $\langle t_{AB} \rangle \approx 1000$ days ≈ 30 months, which close to the reported value (but see below). Similarly one finds $\langle t_{AB} \rangle / \langle t_{CD} \rangle \approx \sqrt{\Omega}$, and that the expected number of CD clones when one AB mutant appears is given by $N_{CD} = N\Omega k^2p^2$ when $N_{AB} = Nk^2p^2 = 1$, and is therefore $N_{CD} = \Omega$. These two predictions exactly fit the scaling laws in fig 3b.

But I think the AB mutation problem can be solved exactly since the accumulation of mutations in each one of the N cells can be viewed as an independent process. For each cell, the probability of acquiring both mutations can be represented by a Markov chain with a state space $\{W, A, B, AB\}$, with transitions $W \rightarrow A$, $W \rightarrow B$, $A \rightarrow AB$, and $B \rightarrow AB$ all occurring with probabilities p (for simplicity I omit the direct transition $W \rightarrow AB$ which occurs with probability p^2 , but its inclusion would be straightforward). The transition matrix for such a Markov chain is $\{\{1 - 2p, p, p, 0\}, \{0, 1 - p, 0, p\}, \{0, 0, 1 - p, p\}, \{0, 0, 0, 1\}\}$ (the AB state is absorbing). The cell starts in the W state (wild-type). Then one can compute exactly the probability that the cell is in the AB state after k days, viz. $\text{Prob}(AB; k) = 1 + (1 - 2p)^k - 2(1 - p)^k$ (for example this can be conveniently done using Mathematica's DiscreteMarkovProcess function). If $p \ll 1$, then $\text{Prob}(AB; k) \approx k(k - 1)p^2$, which is of course to be expected since it is the probability of acquiring the A or B mutation in k days, multiplied by the probability of acquiring the other mutation in the remaining $k - 1$ days. To proceed to the first passage time problem in the ensemble of cells, I note that the probability of a cell NOT being in the AB state after k days is $1 - \text{Prob}(AB; k)$; and consequently the probability of there being no AB mutants in the ensemble of N cells after k days is $[1 - \text{Prob}(AB; k)]^N$. Then, the probability of an AB mutant appearing in the ensemble after k days is $P = 1 - [1 - \text{Prob}(AB; k)]^N$. As far as I can see this is the cumulative distribution function that corresponds to the first passage time problem; and so the problem is solved.

Further progress can be made by taking $\text{Prob}(AB; k) \approx k^2p^2$ which is valid if $k \gg 1$ (but not too large). Then the probability of an AB mutant appearing after k days is $P \approx 1 - (1 - k^2p^2)^N \approx 1 - \exp(-Nk^2p^2)$. With these assumptions, the first passage time distribution itself is $p(k) = \partial P / \partial k =$

$2Nkp^2 \exp(-Nk^2p^2)$. The mean first passage time is then $k_m = \int_0^\infty k p(k) = \sqrt{\pi} / 2p\sqrt{N}$ (which supports the heuristic estimate above), and in terms of k_m one can write $p(k) = (\pi k / 2k_m^2) \exp(-\pi k^2 / 4k_m^2)$. Putting numbers in, and assuming one month is $365 / 12 = 30.4$ days (see above), the mean first passage time $k_m = 886$ days = 29.1 months. Assuming I chose the right number of days in a month, this is in perfect agreement with the result quoted by the author.

Likewise, if I plot the first passage time distribution on top of the AB curve in fig 3a, there appears to be good agreement (modulo normalisation; see above).

An analogous calculation can be made for the CD mutant clones if one neglects the possibility that the Ω -domains overlap. The transition matrix in the CD case would be $\{\{1 - 2p, p, p, 0\}, \{0, 1 - \Omega p, 0, \Omega p\}, \{0, 0, 1 - \Omega p, \Omega p\}, \{0, 0, 0, 1\}\}$. The math goes through as above, with the result that the mean first passage time for the appearance of a CD clone is $k_m = \sqrt{\pi} / 2p\sqrt{(\Omega N)}$ (in agreement with the heuristic argument). Written in terms of this value for k_m , the first passage time distribution is the same as for the AB mutant case.

Some predictions can be made from the above:

First, I would expect a scaling collapse for $\langle k \rangle$ the first passage time distributions shown in fig 3a if the time scale is normalised by the mean first passage time (ie plotted as a function of k / k_m in the above notation). This could easily be tested.

Second, if one takes the cumulative distribution function P , being the probability of an AB mutant (or CD clone) appearing by k days, a plot of $\ln(1 - P)$ against k^2 should be a straight line with slope $-Np^2$ (or $-\Omega Np^2$). This too is easily testable.

Third, the argument indicates that geometry is unimportant, so that the same scaling laws should hold for instance in a 1d linear array of cells. One might test this, but I would not necessarily expect this for the present manuscript.

Finally, from the above analysis it appears that the origin of the $\Omega^{1/2}$ scaling lies in the fact that two mutations are required to form a CD clone, but only the second of these benefits from the 'clonal amplification' effect. In general, if 'n' out of 'm' mutations are 'clonally amplified', then I would expect the scaling to be $\Omega^{n/m}$.

===PREPARING YOUR MANUSCRIPT===

===PREPARING YOUR REVISION IN SCHOLARONE===

<https://royalsociety.org/journals/authors/author-guidelines/#supplementary-material> to include a suitable title and informative caption. An example of appropriate titling and captioning may be found at https://figshare.com/articles/Table_S2_from_Is_there_a_trade-off_between_peak_performance_and_performance_breadth_across_temperatures_for_aerobic_sc_ope_in_teleost_fishes_/3843624.

Author's Response to Decision Letter for (RSOS-201532.R0)

See Appendix A.

RSOS-201532.R1 (Revision)

Review form: Reviewer 1

Is the manuscript scientifically sound in its present form?

Yes

Are the interpretations and conclusions justified by the results?

Yes

Is the language acceptable?

Yes

Do you have any ethical concerns with this paper?

No

Have you any concerns about statistical analyses in this paper?

No

Recommendation?

Accept as is

Comments to the Author(s)

Thank you for taking the time to fully address the issues that I have previously raised regarding this manuscript. I have no further suggestions or criticisms.

Review form: Reviewer 2

Is the manuscript scientifically sound in its present form?

Yes

Are the interpretations and conclusions justified by the results?

Yes

Is the language acceptable?

Yes

Do you have any ethical concerns with this paper?

No

Have you any concerns about statistical analyses in this paper?

No

Recommendation?

Accept with minor revision (please list in comments)

Comments to the Author(s)

The author addressed all the points I made in my previous review. The only outstanding and very minor comment is that the horizontal axis in fig 3d seems mis-annotated. According to the caption this axis is t / t_{Ω} and so should be dimensionless, and also I think the scaled distribution functions should be centered around $t / t_{\Omega} \approx 1$ since $t_{\Omega} = \langle t \rangle$ according to eq 8.

Decision letter (RSOS-201532.R1)

Dear Dr Esposito

On behalf of the Editors, we are pleased to inform you that your Manuscript RSOS-201532.R1 "Cooperation of partially-transformed clones: an invisible force behind the early stages of carcinogenesis" has been accepted for publication in Royal Society Open Science subject to minor revision in accordance with the referees' reports. Please find the referees' comments along with any feedback from the Editors below my signature.

We invite you to respond to the comments and revise your manuscript. Below the referees' and Editors' comments (where applicable) we provide additional requirements. Final acceptance of

your manuscript is dependent on these requirements being met. We provide guidance below to help you prepare your revision.

Please submit your revised manuscript and required files (see below) no later than 7 days from today's (ie 11-Jan-2021) date. Note: the ScholarOne system will 'lock' if submission of the revision is attempted 7 or more days after the deadline. If you do not think you will be able to meet this deadline please contact the editorial office immediately.

on behalf of Prof Catrin Pritchard (Subject Editor)
openscience@royalsociety.org

Associate Editor Comments to Author:

A minor tweak recommended by one of the reviewers, but otherwise the view is that this paper is ready for acceptance - congratulations!

Reviewer comments to Author:

Reviewer: 1

Comments to the Author(s)

Thank you for taking the time to fully address the issues that I have previously raised regarding this manuscript. I have no further suggestions or criticisms.

Reviewer: 2

Comments to the Author(s)

The author addressed all the points I made in my previous review. The only outstanding and very minor comment is that the horizontal axis in fig 3d seems mis-annotated. According to the caption this axis is t / t_{Ω} and so should be dimensionless, and also I think the scaled distribution functions should be centered around $t / t_{\Omega} \approx 1$ since $t_{\Omega} = \langle t \rangle$ according to eq 8.

===PREPARING YOUR MANUSCRIPT===

===PREPARING YOUR REVISION IN SCHOLARONE===

- If you are requesting a discretionary waiver for the article processing charge, the waiver form must be included at this step.
- If you are providing image files for potential cover images, please upload these at this step, and inform the editorial office you have done so. You must hold the copyright to any image provided.
- A copy of your point-by-point response to referees and Editors. This will expedite the preparation of your proof.

- Ensure that your data access statement meets the requirements at <https://royalsociety.org/journals/authors/author-guidelines/#data>. You should ensure that you cite the dataset in your reference list. If you have deposited data etc in the Dryad repository, please only include the 'For publication' link at this stage. You should remove the 'For review' link.
- If you are requesting an article processing charge waiver, you must select the relevant waiver option (if requesting a discretionary waiver, the form should have been uploaded at Step 3 'File upload' above).
- If you have uploaded ESM files, please ensure you follow the guidance at <https://royalsociety.org/journals/authors/author-guidelines/#supplementary-material> to include a suitable title and informative caption. An example of appropriate titling and captioning may be found at https://figshare.com/articles/Table_S2_from_Is_there_a_trade-off_between_peak_performance_and_performance_breadth_across_temperatures_for_aerobic_scope_in_teleost_fishes_/3843624.

Author's Response to Decision Letter for (RSOS-201532.R1)

See Appendix B.

Decision letter (RSOS-201532.R2)

Dear Dr Esposito,

It is a pleasure to accept your manuscript entitled "Cooperation of partially-transformed clones: an invisible force behind the early stages of carcinogenesis" in its current form for publication in Royal Society Open Science.

You can expect to receive a proof of your article in the near future. Please contact the editorial office (openscience@royalsociety.org) and the production office (openscience_proofs@royalsociety.org) to let us know if you are likely to be away from e-mail

contact – if you are going to be away, please nominate a co-author (if available) to manage the proofing process, and ensure they are copied into your email to the journal.

Best regards,

on behalf of Professor Catrin Pritchard (Subject Editor)
openscience@royalsociety.org

Appendix A

Subject Editor Comments to Author (Catrin Pritchard):

Comments to the Author:

Please could the authors address all of the issues raised by both reviewers and we will then reassess the manuscript. I am a little concerned that the second reviewer has recommended extensive changes and I would look to see all of these being addressed in a revised version.

Associate Editor Comments to Author:

Comments to the Author:

While reviewer 1 is largely satisfied with the changes you have made after transfer from JRSI, the second reviewer has extensive queries that we'd like you to address in a revision, please.

I am delighted by the response of the referees. I should note that as a transfer I would have not expected to address new comments particularly on issues of manuscript structure. I had to address what seemed to be opposite suggestions from Ref. 1 in J. Interf. (before transfer) and Ref. 2 in Open Science. As Open Science is not very prescriptive in format, and I find Ref. 2's argument convincing, I have reorganized the manuscript again trying to make good use of the feedback provided by everyone. I have moved details in supporting materials, reorganized the text, and equations. The manuscript should now flow better without interruptions. All information provided in the earlier manuscript is still available as supporting methods, or in the (now shortened) main method sections (section 5).

Ref. 2 also notes that I had not explicitly modelled the $\Omega^{1/2}$ scaling factor shown in (former) Fig. 3b. Their suggestion for a Markov chain model is in fact very welcome particularly because of the level of detail in their feedback that permitted me to elaborate a type of model I did not have experience with. All their suggestion are now integrated in the amended manuscript. I am glad that, as predicted by the referee, the results matched my previous observations, making the work more robust.

In addition to changes to the main manuscript (details below in the point-to-point rebuttal), I also provide the Mathematica notebook I used to analyse the discrete-time Markov chain model, a revised Fig. 3 and simulations, with freely available code (linked to the Github repository <https://doi.org/10.5281/zenodo.4410222>).

Reviewer comments to Author:

Reviewer: 1

Comments to the Author(s)

The author has addressed the vast majority of points that I raised in my review of the original manuscript. However, although they have added additional references related to existing modelling work addressing the interactions of mutations in clones, they have not made it clear how their work sits with regard to these models. I would therefore like to see additional paragraphs added to the Introduction/Background sections to address this important point, clarifying the context of their modelling approach and the novelty/further insight that it provides.

Regarding figure 3 - the grey lines are almost invisible, so I would suggest further revision of this figure to improve visual clarity.

I provide a revised Fig. 3.

I have also integrated further information in the discussion section albeit in a very succinct way because the work is already quite heavy in literature review. The model I propose is rather simple and its purpose is mostly to provoke debate on the role of non-cell-mechanisms during early carcinogenesis. I made even clearer in the amended manuscript this point.

“As computational modelling of multicellular tissues can describe complex homotypic and heterotypic interactions, including short- and long- range interactions, and tissue mechanics [14, 39, 40], computational models rather than analytical tools might be more appropriate to investigate the possible role for ‘oncogenic fields’ in complex mutagenic environments.

The somatic mutation theory is the prevailing model of carcinogenesis which has been described mathematically with several different approaches [11, 13, 18, 41-44]. Modelling work based on evolutionary game theory (e.g., [2, 45, 46]) and analysis of clonal heterogeneity [9], amongst others, have already highlighted the importance of clonal competition and cooperation in cancer. However, mathematical models of somatic mutation theory often do not include cooperation between non- or partially- transformed mutants, particularly when studying the earliest stages of carcinogenesis. Through the lens of the ‘toy model’ we presented here, we show that tissue organisation and cell-to-cell communication might cooperate synergistically with a mutationally driven process particularly during the early stages of carcinogenesis. We emphasise that our work is not in contradiction with prevailing models of oncogenesis, as it is based on similar assumptions but includes explicitly the possibility that non-transformed mutant cells can cooperate. The mathematical analysis we presented was not elaborated to capture more complex phenomena occurring during oncogenesis. However, our analysis suggest that to improve our understanding of carcinogenesis, the identification of the genes and the shared resources that can mediate clonal cooperation - such as growth factors (e.g., mitogens, interleukins, etc.) or metabolic by-products that are often at the basis of cooperative behaviour in lower organisms [47-49] - might be of fundamental importance. ”

Reviewer: 2

Comments to the Author(s)

The manuscript by Esposito examines the role played by cell-cell communication in the early stages of oncogenesis. The postulate is that it is more likely for deleterious mutations to accumulate in a clonal neighborhood than in a single cell, and therefore this route may be relevant for oncogenesis. In itself this suggestion is not surprising since the notion that signaling pathways and inter-cellular communications are relevant to oncogenesis is central at least to some viewpoints of the origins of cancer in somatic cells and tissues. The proposed mechanism is supported by estimates of the size of the relevant neighborhood ($\Omega \approx 10-100$ cells), and Monte-Carlo simulations of a simple model which demonstrates that with all other things being equal, the formation of mutant clones is faster by a factor $\Omega^{1/2}$ (fig 3) compared to the autonomous accumulation of similar mutations in a single cell. Whilst I think the actual importance of clonal co-operation versus cell autonomy must be evaluated on a case-by-case basis, I do accept the argument by the author that the models presented here make the point in quite a clear and dramatic manner. Therefore I support publication on the grounds of general interest, with revision to take into account the comments below.

I would like to thank the referee for their support and advice.

The material seems a little disjointed. I recognize there is a desire to partition the presentation into ‘materials and methods’ and ‘results’, but in this case it seems to introduce unhelpful and artificial barriers to the natural progression of the argument. It might be better to collect together all the material for the Ω estimates and present it first, as a complete whole, before moving on to presenting the Monte-Carlo simulations. In general, I found the presentation a bit labored and over-

elaborated in places. For example the first two equations in section 3.2 could easily be combined (the sum is trivial). Likewise the presentation in sections 3.3 and 3.4 could be greatly condensed.

I agree. The manuscript was originally organized with the maths presented in an Appendix and the Appendix was designed to guide those readers who might be mathematically trained but not using maths routinely. As the Royal Society Open Science journal does not have strict formatting rules, I therefore propose to move the methods section at the bottom of the manuscript. A succinct description of the maths is included in the Result section, with the lengthier descriptions moved as supporting methods.

This strategy should address the referee’s concern still ensuring that readers from different backgrounds will have sufficient information to study the proposed model. All changes are tracked in the submission and are too substantial to summarize here.

In fig 2 and the accompanying discussion in section 3.6, I think this is not really a comparison between ‘analytic’ and ‘numerical’ results as such, but rather an assessment of the effect of truncating the sums. I don’t think anyone doubts the ability of MATLAB or another numerical platform to evaluate the truncated sums, so the benchmark ‘analytic’ results would be better stated as results for $k_{max} \rightarrow \infty$ (fig 2 annotation), in my opinion.

I have amended Fig. 2.

However, the purpose of the comparison was two-fold:

- 1) Confirming the validity of Eq. 4 ($\Omega = 12\zeta(l - 1) - 8\zeta(l) + 4/3$) as analytical solution of the problem
- 2) Testing if the general discussion of the solution held also when some assumptions were not satisfied (e.g. a finite tissue).

I made this clearer also in the amended figure legend.

“Figure 2. Numerical validation of the oncogenic field solutions. a) Comparison between the numerical evaluation of Eq. 2 for finite values of k_{max} and the estimates of the oncogenic field factor Ω obtained by the analytical representations for $k_{max} \rightarrow \infty$ for a three-dimensional (3D, Eq. 3) and three-layered (TL, Eq. 4) tissue model. The solid lines represent the analytical solutions within the limits of its convergence ($l > 2$ for TL in blue and $l > 3$ for 3D in red). The results evaluated over smaller (empty circles) and larger (solid circles) neighbourhood with 10 and 100 cell radii, respectively, represent cases where the assumption of a tissue of infinite extension used to evaluate (Eqs. 3-4) is not valid. **b)** Identical comparisons as described in a) but for the exponential decay model for

a three-dimensional tissue. Both the analytical solution (Eq. 5, $k_{max} \rightarrow \infty$) and numerical estimates of the finite series (Eq. 2, $k_{max}=10, 100$) converge to the value of $\Omega \sim 8$ for steep decays. c) Values for Ω computed for a general case where the oncogenic field decays jointly as the inverse of a power-law and exponentially (Eq. 6). Eq. 6 (solid lines) is compared to the finite sums ($k_{max}=10, 100$) for the same parameter sweep shown in a) and b), i.e. with the inverse power from 1 to 10 and with a decay constant k_c from 0.1 to 10.”

In section 4.1 there is a statement “The formalism for exponentially decaying oncogenic fields is less elegant”. I presume the author means the expressions for pure power-law decays are more succinct? However, there is a much better biophysical motivation for exponential decay (it results from the competition between diffusion and degradation), or for exponentially-truncated power-laws, so I don’t see how elegance (or succinctness) can be a relevant criterion here (cf “elegance is for tailors”, attributed to Boltzmann). I have removed that statement. Elegant was used to state that the equation was interpretable in a more intuitive way and therefore was very instructive. However, that statement might have been ambiguous and certainly was not necessary.

Please provide error estimates if available for the mean first passage times quoted in section 4.2 (ie, 29 months for first appearance of double mutant cells; and 14.7, 10.1, 8.5 and 6.2 months respectively for $\Omega = 4, 8, 12$ and 24, for the appearance of mutant clones). Also, for clarity, please define ‘months’ (and ‘years’) in terms of the number of days (see also below).

To avoid any ambiguity, the results are now presented only in simulation years and the year is explicitly defined as a 365 days long at the first occurrence in the text in the amended manuscript. I have rerun identical simulations five times (each still comprising 2000 repeats) to provide error estimates. The amended text is:

“The average time for a double-mutant cell to appear ($\langle t_{AB} \rangle$) is 2.43 ± 0.04 simulation years (mean \pm standard deviation computed over 5 independent Monte Carlo simulations each made of 2,000 runs with 1 year defined as 365 simulation days). The distribution of $\langle t_{AB} \rangle$ values depends only on ρ_0 and N but not on Ω and, therefore, we show the average of the four distributions of $\langle t_{AB} \rangle$ values (Fig. 3a, black curve) as reference for the cooperating mutants. The distribution of the waiting times for the appearance of CD clones depends on the value of Ω (Fig. 3a, coloured curve) and exhibits an average time ($\langle t_{CD} \rangle$) of about 1.23 ± 0.02 , 0.86 ± 0.03 , 0.70 ± 0.06 and 0.50 ± 0.04 months for Ω values equal to 4, 8, 12 and 24, respectively – scaling as $\Omega^{-0.5}$ (Fig. 3c).”

I did not understand the normalisation being used in fig 3a. If the vertical axis is frequency ‘per year’, then I think these curves are all too low. For example the support for the $\Omega = 24$ curve is almost all concentrated in the first year, so wouldn’t one expect the peak to be $> O(1)$ in units of years^{-1} (the integrated area under the curve should be unity, for a probability distribution). Also, when I try to compare the curves with the analytic results derived below, they all seem too low by a constant factor.

The original PDF was histogrammed with a resolution of 32 bins over the plotted 6 years. The units of the ordinates were thus $\sim 0.44 \text{months}^{-1}$. I had not noticed this discrepancy and the factor 0.44 would have accounted for the differences highlighted. I had resampled the distributions and explicitly reported the units (1/month, see Fig. 3a).

To facilitate reproducibility and comparison from future readers, I have now added to the supporting files not just the code used but also the output of the Monte Carlo simulations I use to generate the figures.

Please number all displayed equations! This is Fisher's rule – see “What's Wrong with these Equations?”, N. D. Mermin, Physics Today 42, 9 (1989) [a PDF can be found by searching for the title].

[Also there are no page numbers in the manuscript, which makes it harder to refer to the text.]

The manuscript was amended according to the referee's suggestions.

The main remaining problem I have with the manuscript as it stands is that there is no proper discussion that I could see of the possible origin of the $\Omega^{1/2}$ scaling result, nor is there any attempt to explain the apparently exact scaling laws in fig 3b. But I note that in the Monte-Carlo model the appearance of AB mutant cells or CD mutant clones is essentially a first passage time problem [see for example N. van Kampen, Stochastic Processes in Physics and Chemistry (North-Holland, 1981)]. As such, I think the results are fairly easy to rationalize with a modicum of statistical insight. I first give a heuristic analysis and then present what I think is likely to be an exact solution to the AB mutant cell problem, and a good approximation for the CD mutant clone case. From these one can make some predictions which could be tested. The inclusion of these tests, supported by some version of the arguments made below, would I think strengthen the manuscript.

Given that the mutations occur with probability $p \ll 1$ per day, after k days (with kp small) the probability that a cell will acquire an A mutation is $\approx kp$, and the probability that it will acquire a B mutation is likewise. Therefore in a population of size N , the expected number of AB mutants will be $N_{AB} \approx Nk^2p^2$. Likewise the probability that a cell will acquire a C mutation is $\approx kp$, and the probability that a cell in its neighborhood of size Ω will acquire a D mutation is $\approx \Omega kp$, and therefore the expected number of CD mutant clones will $N_{CD} \approx N\Omega k^2p^2$. From these one can estimate the mean first passage times as the number of days for one AB mutant cell or one CD mutant clone to appear; these being respectively $\langle t_{AB} \rangle \approx 1 / pvN$ and $\langle t_{CD} \rangle \approx 1 / pv(\Omega N)$. For example with $p = 10^{-6}$ and $N = 10^6$, one has $\langle t_{AB} \rangle \approx 1000$ days ≈ 30 months, which close to the reported value (but see below). Similarly one finds $\langle t_{AB} \rangle / \langle t_{CD} \rangle \approx \sqrt{\Omega}$, and that the expected number of CD clones when one AB mutant appears is given by $N_{CD} = N\Omega k^2p^2$ when $N_{AB} = Nk^2p^2 = 1$, and is therefore $N_{CD} = \Omega$. These two predictions exactly fit the scaling laws in fig 3b.

But I think the AB mutation problem can be solved exactly since the accumulation of mutations in each one of the N cells can be viewed as an independent process. For each cell, the probability of acquiring both mutations can be represented by a Markov chain with a state space $\{W, A, B, AB\}$, with transitions $W \rightarrow A$, $W \rightarrow B$, $A \rightarrow AB$, and $B \rightarrow AB$ all occurring with probabilities p (for simplicity I omit the direct transition $W \rightarrow AB$ which occurs with probability p^2 , but its inclusion would be straightforward). The transition matrix for such a Markov chain is $\{\{1 - 2p, p, p, 0\}, \{0, 1 - p, 0, p\}, \{0, 0, 1 - p, p\}, \{0, 0, 0, 1\}\}$ (the AB state is absorbing). The cell starts in the W state (wild-type). Then one can compute exactly the probability that the cell is in the AB state after k days, viz. $\text{Prob}(AB; k) = 1 + (1 - 2p)^k - 2(1 - p)^k$ (for example this can be conveniently done using Mathematica's DiscreteMarkovProcess function). If $p \ll 1$, then $\text{Prob}(AB; k) \approx k(k - 1)p^2$, which is of course to be expected since it is the probability of acquiring the A or B mutation in k days, multiplied by the probability of acquiring the other mutation in the remaining $k - 1$ days. To proceed to the first

passage time problem in the ensemble of cells, I note that the probability of a cell NOT being in the AB state after k days is $1 - \text{Prob}(\text{AB}; k)$; and consequently the probability of there being no AB mutants in the ensemble of N cells after k days is $[1 - \text{Prob}(\text{AB}; k)]^N$. Then, the probability of an AB mutant appearing in the ensemble after k days is $P = 1 - [1 - \text{Prob}(\text{AB}; k)]^N$. As far as I can see this is the cumulative distribution function that corresponds to the first passage time problem; and so the problem is solved.

Further progress can be made by taking $\text{Prob}(\text{AB}; k) \approx k^2 p^2$ which is valid if $k \gg 1$ (but not too large). Then the probability of an AB mutant appearing after k days is $P \approx 1 - (1 - k^2 p^2)^N \approx 1 - \exp(-Nk^2 p^2)$. With these assumptions, the first passage time distribution itself is $p(k) = \partial P / \partial k = 2Nkp^2 \exp(-Nk^2 p^2)$. The mean first passage time is then $k_m = \int_0^\infty dk k p(k) = \sqrt{\pi} / 2pvN$ (which supports the heuristic estimate above), and in terms of k_m one can write $p(k) = (\pi k / 2k_m^2) \exp(-\pi k^2 / 4k_m^2)$. Putting numbers in, and assuming one month is $365 / 12 = 30.4$ days (see above), the mean first passage time $k_m = 886$ days = 29.1 months. Assuming I chose the right number of days in a month, this is in perfect agreement with the result quoted by the author. Likewise, if I plot the first passage time distribution on top of the AB curve in fig 3a, there appears to be good agreement (modulo normalisation; see above).

An analogous calculation can be made for the CD mutant clones if one neglects the possibility that the Ω -domains overlap. The transition matrix in the CD case would be $\{\{1 - 2p, p, p, 0\}, \{0, 1 - \Omega p, 0, \Omega p\}, \{0, 0, 1 - \Omega p, \Omega p\}, \{0, 0, 0, 1\}\}$. The math goes through as above, with the result that the mean first passage time for the appearance of a CD clone is $k_m = \sqrt{\pi} / 2pv(\Omega N)$ (in agreement with the heuristic argument). Written in terms of this value for k_m , the first passage time distribution is the same as for the AB mutant case.

Some predictions can be made from the above:

First, I would expect a scaling collapse for *all* the first passage time distributions shown in fig 3a if the time scale is normalised by the mean first passage time (ie plotted as a function of k / k_m in the above notation). This could easily be tested.

Second, if one takes the cumulative distribution function P , being the probability of an AB mutant (or CD clone) appearing by k days, a plot of $\ln(1 - P)$ against k^2 should be a straight line with slope $-Np^2$ (or $-\Omega Np^2$). This too is easily testable.

I genuinely grateful for the time invested in providing their advice. In the original manuscript I had mentioned just that “One cell accrues pairs of mutations at the rate ρ_0 but within a neighbourhood cooperating cells accrue mutations at an apparent rate of $\rho_0 \sqrt{\Omega}$. We tested this simple mathematical inference with Monte Carlo simulations (see **Fig. 3** and **Section 3.6**.” I left as an implicit consequence that the scaling factors was the direct consequence of that apparent rate as described in the ‘heuristic explanation’ provided by the referee.

However, not only I recognize the need to be more explicit in explaining this scaling factor but I appreciate the detailed suggestion of the Markov Chain model. I have amended the manuscript to integrate a brief analysis of the ‘first passage problem’ both in the main text and supplementary materials, which now include an annotated Mathematica notebook as well.

The section I added in the main text is:

“Next, to better explain and generalise the origin of the scaling factor $\Omega^{-0.5}$, we modelled the mutational process as a discrete-time Markov chain [16, 17] (see **Section S.5** in **Sup. Methods** and **Sup. Files**). We show that for a ‘two-hits’ model, the distributions shown in **Fig. 3a** can be described analytically as:

$$p_{AB}(t) \approx 2tN\rho_0^2 e^{-t^2 N\rho_0^2} \quad \text{Eq. 7a}$$

$$p_{CD}(t) \approx 2tN\rho_0^2 \Omega e^{-t^2 N\rho_0^2 \Omega} \quad \text{Eq. 7b}$$

Fig. 3d shows the very good match between the Monte Carlo simulations and the Markov chain model. Using **Eqs. 7**, we can then estimate the cell-autonomous time-horizon as the average time required to observe the first AB-clone in N cells (**Eq. 8a**) and, similarly, the average latency to observe the first cooperating CD-clone (**Eq. 8b**).

$$t_a = \langle t_{AB} \rangle \cong \frac{1}{2\rho_0} \sqrt{\frac{\pi}{N}} \quad \text{Eq. 8a}$$

$$t_\Omega = \langle t_{CD} \rangle \cong \frac{1}{2\rho_0} \sqrt{\frac{\pi}{N\Omega}} = t_a \Omega^{-0.5} \quad \text{Eq. 8b}$$

t_Ω is indeed rescaled by a factor $\Omega^{-0.5}$ relative to t_a as observed in the Monte Carlo simulations. **Eqs. 8** provides estimates for $t_a=2.43$ years and $t_\Omega = 1.21, 0.86, 0.70$ and 0.50 years (for $\Omega=, 4, 8, 12,$ and $24,$ respectively), values that are in excellent agreement with the numerical simulations. We note that for the more general case where m mutations cooperates through non-cell-autonomous mechanisms, the scaling factor $\Omega^{-0.5}$ would assume the form $\Omega^{-d/m}$ shown in **Eq. S.5.9**, with d representing the number of mutations that cooperate at distance defined by the parameter Ω . For example, in the case where a mutant cell C interacts with a mutant cell D via paracrine effects and D reciprocates, the scaling factor is Ω ($d=m=2$). However, if a C-mutant with $m-1$ mutations benefits from a D-mutant (with a single mutation) in its neighbourhood, this scaling factor is $\Omega^{-1/m}$. “

The new results are also shown in **Fig. 3d**. I compare the results obtained with the Monte Carlo simulations to the shape of the distribution (**Eq. 8**) predicted by the Markov Chain model. I plotted this data on a new time base defined by k/km as per suggestion (t/t_Ω in my formalism). As predicted by the referee, the distributions for different Omega values collapse to identical distributions described by **Eqs. 8**. Having directly compared the distributions, I did not plot ‘ $\ln(1 - P)$ ’ against k^2 ’ as this would have been redundant.

The amended figure and legend are:

Figure 3. Monte Carlo simulations of the cell-autonomous time-horizon. **a)** Probability distribution of the waiting-times for the occurrence of the first two co-occurring mutations (AB-clones, black curve) or for the first cooperating mutations (CD-clones, coloured curves) through non-cell-autonomous mechanisms for Ω values equal to 4, 8, 12 and 24. The coloured curves are the average of the five independent simulations runs (see **Methods**). The black curve is the average of twenty (4 values of $\Omega \times 5$ repeats) runs as each simulation had its own AB-control. **b)** Distribution (normalised to maximum for better visualisation) of the number of CD-clones at the end of the simulations ($t=t_{end}$). t_{end} is the time at which at least one AB- and one CD- clone are detected. **c)** The average time ($\langle t_{CD} \rangle$) at which the first cooperating CD-clone is observed scales with the square root of Ω (red line) compared to the average time ($\langle t_{AB} \rangle$) at which the first AB-mutant appears. The average number of cooperating mutations within a neighbourhood at $t=t_{end}$ ($\langle N_{CD} \rangle$) scales as Ω (blue curve). Errors are standard deviations for five independent sets of Monte Carlo simulations. **d)** Distribution of the waiting-times for CD-clones as shown in panel a but replotted on a new time-base defined, for each curve, as t/t_{Ω} . t_{Ω} was predicted by the Markov Chain model using Eq. 8b. The distributions predicted by the Markov Chain model (Eqs. 7, black curves) fully overlap with the result of the numerical simulations (circles). For better comparison, the distributions were normalized to the sum and offset with a constant.

Third, the argument indicates that geometry is unimportant, so that the same scaling laws should hold for instance in a 1d linear array of cells. One might test this, but I would not necessarily expect this for the present manuscript.

Finally, from the above analysis it appears that the origin of the $\Omega^{1/2}$ scaling lies in the fact that two mutations are required to form a CD clone, but only the second of these benefits from the 'clonal amplification' effect. In general, if 'n' out of 'm' mutations are 'clonally amplified', then I would expect the scaling to be $\Omega^{n/m}$.

As advised I have rather focused on the latter suggestion (see response to previous point). I have included a detail description of the Markov Chain model and its prediction in a Mathematica notebook shared with this submission.

===PREPARING YOUR MANUSCRIPT===

- one version identifying all the changes that have been made (for instance, in coloured highlight, in bold text, or tracked changes);
- a 'clean' version of the new manuscript that incorporates the changes made, but does not highlight them. This version will be used for typesetting if your manuscript is accepted.

While not essential, it will speed up the preparation of your manuscript proof if accepted if you format your references/bibliography in Vancouver style (please see <https://royalsociety.org/journals/authors/author-guidelines/#formatting>). You should include DOIs

for as many of the references as possible.

===PREPARING YOUR REVISION IN SCHOLARONE===

-- If you have uploaded ESM files, please ensure you follow the guidance at <https://royalsociety.org/journals/authors/author-guidelines/#supplementary-material> to include a suitable title and informative caption. An example of appropriate titling and captioning may be found at https://figshare.com/articles/Table_S2_from_Is_there_a_trade-off_between_peak_performance_and_performance_breadth_across_temperatures_for_aerobic_scops_in_teleost_fishes_/3843624.

Journal Name: Royal Society Open Science

Journal Code: RSOS

Online ISSN: 2054-5703

Journal Admin Email: openscience@royalsociety.org

Journal Editor: Andrew Dunn

Journal Editor Email: openscience@royalsociety.org

MS Reference Number: RSOS-201532

Article Status: SUBMITTED

MS Dryad ID: RSOS-201532

MS Title: Cooperation of partially-transformed clones: an invisible force behind the early stages of carcinogenesis

MS Authors: Esposito, Alessandro

Contact Author: Alessandro Esposito

Contact Author Email: ae275@cam.ac.uk

Contact Author Address 1: New Museums Site

Contact Author Address 2: Pembroke Street

Contact Author Address 3:

Contact Author City: Cambridge

Contact Author State:

Contact Author Country: United Kingdom of Great Britain and Northern Ireland

Contact Author ZIP/Postal Code: CB23RA

Keywords: oncogenesis, modelling, non-cell-autonomous

Abstract: Most tumours exhibit significant heterogeneity and are best described as communities of cellular populations competing for resources. Growing experimental evidence also suggests, however, that cooperation between cancer clones is important as well for the maintenance of tumour heterogeneity and tumour progression. However, a role for cell communication during the earliest steps in oncogenesis is not well characterised despite its vital importance in normal tissue and clinically manifest tumours. Here, we present a simple analytical model and stochastic lattice-based simulations to study how the interaction between the mutational process and cell-to-cell communication in three-dimensional tissue architecture might contribute to shape early oncogenesis. We show that non-cell-autonomous mechanisms of carcinogenesis could support and accelerate pre-cancerous clonal expansion through the cooperation of different, non- or partially-transformed mutants. We predict the existence of a 'cell-autonomous time-horizon', a time before which cooperation between cell-to-cell communication and DNA mutations might be one of the

most fundamental forces shaping the early stages of oncogenesis. The understanding of this process could shed new light on the mechanisms leading to clinically manifest cancers.

EndDryadContent

Appendix B

Dear Prof Pritchard and colleagues,

I am delighted by the peer-review process's outcome, and I would like to thank you for handling the manuscript. Because I was asked only to change an axis's labelling, I do not provide a tracked version of the manuscript. The only change I have applied is to Fig. 3d as per referee 2's advice, i.e. I have relabelled Fig. 3d as t/t_{ω} . I have also modified the correspondent m-files provided in the EMS to annotate the change of time bases on the axis.

Kind regards,

Alessandro

Dr Alessandro Esposito
Group Leader

MRC Cancer Unit, University of Cambridge Hutchison/MRC Research Centre Box 197, Biomedical
Campus Cambridge, United Kingdom
CB2 0XZ

+44(0)1223 330605 (office)

+44(0)1223 763241 (FAX)

+44(0)795 4383731 (mobile)

quantitative-biology.org (group page)

OncoLive Project website

MRC Cancer Unit (institutional homepage)